# 3D-Printed HA-Based Scaffolds for Bone Regeneration: Microporosity, Osteoconduction and Osteoclastic Resorption

**DOI:** 10.3390/ma15041433

**Published:** 2022-02-15

**Authors:** Chafik Ghayor, Indranil Bhattacharya, Julien Guerrero, Mutlu Özcan, Franz E. Weber

**Affiliations:** 1Center of Dental Medicine, Oral Biotechnology & Bioengineering, University of Zurich, Plattenstrasse 11, 8032 Zurich, Switzerland; chafik.ghayor@usz.ch (C.G.); indranil.bhattacharya@usz.ch (I.B.); julien.guerrero@usz.ch (J.G.); 2Center of Dental Medicine, Division of Dental Biomaterials, Clinic for Reconstructive Dentistry, University of Zurich, Plattenstrasse 11, 8032 Zurich, Switzerland; mutlu.ozcan@zzm.uzh.ch; 3CABMM, Center for Applied Biotechnology and Molecular Medicine, University of Zurich, Winterthurerstr. 190, 8057 Zurich, Switzerland

**Keywords:** hydroxyapatite, microporosity, osteoconduction, macroarchitecture, microarchitecture, nanoarchitecture, bone substitute, additive manufacturing, 3D printing, ceramics, tricalcium phosphate

## Abstract

Additive manufacturing enables the realization of the macro- and microarchitecture of bone substitutes. The macroarchitecture is determined by the bone defect and its shape makes the implant patient specific. The preset distribution of the 3D-printed material in the macroarchitecture defines the microarchitecture. At the lower scale, the nanoarchitecture of 3D-printed scaffolds is dependent on the post-processing methodology such as the sintering temperature. However, the role of microarchitecture and nanoarchitecture of scaffolds for osteoconduction is still elusive. To address these aspects in more detail, we produced lithography-based osteoconductive scaffolds from hydroxyapatite (HA) of identical macro- and microarchitecture and varied their nanoarchitecture, such as microporosity, by increasing the maximum sintering temperatures from 1100 to 1400 °C. The different scaffold types were characterized for microporosity, compression strength, and nanoarchitecture. The in vivo results, based on a rabbit calvarial defect model showed that bony ingrowth, as a measure of osteoconduction, was independent from scaffold’s microporosity. The same applies to in vitro osteoclastic resorbability, since on all tested scaffold types, osteoclasts formed on their surfaces and resorption pits upon exposure to mature osteoclasts were visible. Thus, for wide-open porous HA-based scaffolds, a low degree of microporosity and high mechanical strength yield optimal osteoconduction and creeping substitution. Based on our study, non-unions, the major complication during demanding bone regeneration procedures, could be prevented.

## 1. Introduction

Bone tissue engineering has emerged from the need to satisfy the current unmet demand of bone grafts and to alleviate the problems associated with autografts and allografts [1]. It focuses on methods to synthesize and/or regenerate bone to restore, maintain, or improve its functions in vivo [2]. As for all tissue engineering specialties, bone tissue engineering can comprise materials, growth factors, and cells [3]. Such a combination can, for example, be realized by additive manufacturing with a synergistic combination of materials, growth factors, and cells, as defined previously [4].

Additive manufacturing is a methodology to build objects layer-by-layer. This not only allows the spatial defined distribution of material, growth factors, and cells but is a key technology for personalized medicine realized by diverse methodologies [5]. Three-dimensional printing for the production of bone substitutes has been reviewed extensively [4,6,7,8]. The architecture of such a personalized bone substitute is defined on three levels. The first level, namely the macroarchitecture, is given by the shape of the bone defect and defines the centimeter-to millimeter-sized outer shape of a scaffold. The second level, the microarchitecture determines the spatial distribution of the material inside the macroarchitecture, defining pore size, porosity, channels, and pore interconnectivity from millimeter to micrometer size. The macro- and microarchitecture are encrypted in the Standard Triangle Language (stl)-file used to program the 3D printer. The third level of the architecture of a 3D-printed scaffold, the nanoarchitecture, depends on the post-processing methodology such as sintering regimes, surface treatments, and to a lesser extent on the underlying additive manufacturing procedures used today [1,9]. Moreover, all three levels are well known to be critical for osteoconduction [9,10].

Osteoconduction is a process of ingrowth of capillaries, perivascular tissue, and osteoprogenitor cells from a bony bed into the 3D structure of a porous implant [11,12] used as a guiding cue to bridge a defect with bony tissue [9]. Bone tissue ingrowth to repair a bony defect depends heavily on the microarchitecture. With the introduction of additive manufacturing to the field, the former recommendations for microarchitectural features, such as pore size could be tested [9]. In the 1990s, the optimal pore size in bone substitutes was set to a pore diameter of 0.3 to 0.5 mm [13,14,15,16]. One in vivo study with different scaffolds produced conventionally by salt leaching reported that bone ingrowth was independent of pore diameters from 0.15 up to 1.22 mm [17]. More recently and based on a library of highly defined scaffolds produced by additive manufacturing, an in-depth reevaluation of osteoconductive microarchitectures revealed that pores of 1.2 mm were superior in terms of osteoconduction; the results were defined by bone ingrowth velocity as compared with pores of 0.5 mm or 1.5 and 1.7 mm [18].

The presence of cells with micropores well below 0.1 mm in diameter and bone ingrowth have been reported by several groups [19,20,21]. It is important to note that the definition of microporosity with pore diameters below 50 µm, as used in bone tissue engineering [22], is not in line with IUPAC nomenclature [23] for micropores, which discriminates between pore diameters below 2 nm, between 2 and 50 nm, and more than 50 nm. Variations of the percentage of microporosity in tricalcium phosphate (TCP)-based scaffolds, in the sense of bone tissue engineering, have no effect on bone regeneration and osteoconductivity [24]. Recent results from our group, however, showed that for wide-open porous, highly osteoconductive microarchitectures from TCP, a high microporosity of 39% significantly enhanced osteoconductivity [10]. Since we suspected an indirect link between microporosity and osteoconductivity reflecting differences in ion dissolution, here, we extended our studies to hydroxyapatite (HA)-based scaffolds known for their slower ion dissolution rate as compared with TCP [25]. To study this in more detail, we produced HA-based scaffolds of identical, wide-open microarchitectures through additive manufacturing known to be osteoconductive and varied their microporosity by increasing the maximal sintering temperature during post-processing. Our aim was to evaluate the contribution of microporosity to osteoconductivity and the osteoclastic resorbability of wide-open porous HA-based scaffolds.

## 2. Materials and Methods

### 2.1. Implant Production

The macro- and microarchitecture of the scaffolds from HA were identical to the one that we used to study the effect of microporosity in TCP-based scaffolds [10]. In essence, the cuboidal 1.5 × 1.5 × 1.5 mm^3^ unit cell used to assemble a scaffold holds a 1.2 mm pore in the center. This pore is connected to all six sides of the unit cell by centrally located cylinders with a diameter of 0.7 mm.

The HA slurry LithaBone™ HA 400 (Lithoz, Vienna, Austria) was used to build the scaffolds [26]. In brief: The scaffolds were built by layers of 25 µm thickness at a resolution of 50 µm in the x/y-plane using a CeraFab 7500 system (Lithoz, Vienna, Austria). The slurry of each layer solidified at defined locations by the exposure of the photoactive polymer to a blue LED light [27]. After production, the so formed green body was cut from the building platform by a razor blade, and cleaned with LithaSol 30™ (Lithoz, Vienna, Austria) and pressurized air. The polymeric binder in the green bodies was decomposed during a thermal treatment regime. The remaining ceramic particles were densified to different degrees by sintering with a dwelling time of 2 h at 1100, 1200, 1300, or 1400 °C. The exposure of the green bodies to the different sintering temperatures let them shrink differently. These differences were compensated for by adjusting all three dimensions to yield scaffolds of identical macro- and microarchitecture.

### 2.2. Scanning Electron Microscopy (SEM)

The scaffolds were analyzed by a service lab at the University of Zurich using a Zeiss Supra V50 scanning electron microscope (SEM) (Carl Zeiss, Oberkochen, Germany). Scanning occurred under an acceleration voltage of 12 kV with a distance between the sample and the detector of 9.5 cm.

### 2.3. Microporosity

The shrinkage parameters in all three dimensions used to compensate for the shrinkage of the green bodies exposed to this array of dwelling temperatures during post-processing were used to calculate microporosity. Microporosity of 0% was set at samples sintered at 1400 °C. Experimentally, microporosity was measured from the amount of distilled water taken up by the test scaffolds. The test scaffolds were set together by using an upper circular platform (5.0 mm in diameter and 2.5 mm thickness) and a lower circular platform (4.0 mm in diameter and 1.5 mm thickness).

### 2.4. Compression Strength Measurements

Cuboids of 7.5 × 7.5 × 6.0 mm were composed of cuboidal 1.5 × 1.5 × 1.5 mm^3^ unit cells with a 1.2 mm pore in the center and 0.7 mm connections. The specimens were mounted in the jig of a universal testing machine (Zwick ROELL Z2.5 MA 18-1-3/7, Ulm, Germany). An area of 6.0 mm × 7.5 mm, in a direction perpendicular to the building layers, was subjected to compressive loading at a crosshead speed of 1 mm/min. The software program (TestXpert V11.02, Zwick ROELL, Ulm, Germany) was used to determine the maximal compression strength.

### 2.5. Ion Release

The Ca^2+^ release from HA scaffolds was determined in ddH_2_O at 37 °C, as previously reported [10]. In brief: 250 mg scaffolds were placed in a 5 mL tube (Eppendorf) containing 2.5 mL ddH_2_O. Calcium release was measured using a Quantichrom Calcium Assay Kit DICA-500 (BioAssay Systems, Hayward, CA, USA) from 500 μL aliquots. The aliquot was replaced by 500 μL of fresh ddH_2_O.

### 2.6. Specific Surface Evaluation

The BET methodology [28] via the absorption of N_2_ in an SA 3100 Surface Area and Pore Volume analyzer (Beckman Coulter, Orange County, CA, USA). Prior to the analysis performed at −196 °C from 0.05 to 0.2 bar, the samples were dried for 2 h at 180 °C.

### 2.7. Surgical Procedure

Osteoconductivity of scaffolds was assessed in vivo with ten adult New Zealand White rabbits in a calvarial defect model, as reported earlier [29]. The procedure was evaluated and accepted by the local authorities (065/2018). Sample handling was reported in [30].

### 2.8. Histomorphometry

Histomorphometry was performed based on the ground section from the middle of each implant using image analysis software (Image-Pro Plus^®^, Media Cybernetic, Silver Springs, MD, USA), as reported earlier [10].

### 2.9. Bone Bridging

The determination of bony bridging as a measure for osteoconduction was performed as reported earlier [31,32].

### 2.10. Osteoclast Differentiation on HA Scaffold and Resorption Pit Assay

The RAW264.7 cells were cultured, as previously reported [10]. The morphology of the cells and resorption lacuna were evaluated by scanning electron microscope (SEM). For cells, samples were rinsed with 0.1 M phosphate-buffered saline (PBS) and fixed with 2.5% glutaraldehyde solution (SIGMA) overnight at 4 °C, dehydrated and dried. After gold coating, cell morphology could be studied in the SEM. For the resorption lacuna detection, cells were removed from the surface by NaOCl solution. Control discs were generated with cells but without RANKL supplement.

### 2.11. Statistics

Statistics was performed as previously reported [10]. Where appropriate, the Jonckheere–Terpstra test was applied. Values are reported in the text by mean ± standard deviation or displayed in graphs as median ± lower/upper quartile.

## 3. Results

### 3.1. Scaffold Characterization

After sintering, the scaffolds appeared bluish, with an increase in color intensity, in line with the sintering temperature (Figure 1a). According to the manufacturer of the slurry, the blue color is the effect of traces of manganese in the HA particles. It appears after high-temperature sintering in an oxidizing atmosphere [33].

HA grains grew depending on the sintering temperature from 1.07 ± 0.46 µm at 1100 °C to 5.98 ± 1.11 µm at 1400 °C and underwent partial fusion (Figure 1b). The micropore diameter increased from 1.17 ± 0.47 µm at 1100 °C to 1.67 ± 0.52 µm at 1400 °C. Moreover, surface and microporosity was affected by partial sintering (Table 1). The surface decreased from 0.79 m^2^/g at 1100 °C to 0.24 m^2^/g at 1400 °C and microporosity from 45.85 ± 0.39% to 0.74 ± 1.87%, respectively.

### 3.2. Microporosity of HA-Based Scaffolds

The shrinkage compensation in all three axes was used to determine the percentage of microporosity in the material. In addition, microporosity was experimentally determined by weight gain due to infiltration with distilled water. The percentages of microporosity determined by both approaches (Table 1) were almost identical. Therefore, H_2_O could fill up the entire microporosity system of the test samples to a depth of 4.0 mm within minutes. The maximal microporosity determined by infiltration of samples sintered at 1000 °C with water was 46.43 ± 0.35%. According to the manufacturer, the volume percentage of the binder in the slurry is 54%. That means that at least 86% if not close to 100% of the microporosity is open porous. Moreover, if the volume of HA in the scaffolds is calculated according to its specific weight and the microporosity determined by the volume of water infiltration as before, the numbers match the numbers displayed in Table 1. Therefore, due to the homogeneous dispersion of the HA particles in the binder system, debinding yields the formation of an almost exclusive open-porous microporosity where water and water-soluble substances can easily pass through.

### 3.3. Osteoconductivity of Microporous HA-Scaffolds In Vivo

The microarchitecture of all scaffolds was identical as they were built with the same “stl-file” and compensated by using the printer software for x, y, and z-dimension induced sintering temperature-dependent shrinkage (Figure 1a). Scaffolds varied in grain size, surface, and microporosity (Table 1). The histologies of the middle sections (Figure 2a) revealed that over 4 weeks of implantation, sintering temperature variations between 1100 and 1400 °C had no significant effect on osteoconductivity, determined by the degree of defect bridging (Figure 2b). For the extent of bony regenerated area (Figure 2c), significant changes due to the variation of sintering temperature could not be detected, although a trend was observed, i.e., higher sintering temperatures induced a slight reduction in this measure.

As measures, bony bridging and the degree of bony regenerated area in the defect were determined (Figure 2b,c, respectively). In defects treated with scaffolds sintered at 1100 °C, 83.62 ± 22.77% of the middle section was bridged, 90.67 ± 10.20% for those sintered at 1200 °C, 89.25 ± 16.06% for those sintered at 1300 °C, and 70.36 ± 30.75% for those sintered at 1400 °C, respectively. In the area of interest, the percentage of bony regeneration in the middle section with scaffolds sintered at 1100 °C was 67.61 ± 24.60%, 69.84 ± 16.32% for 1200 °C sintered scaffolds, 63.73 ± 23.21% for 1300 °C sintered scaffolds, and 50.55 ± 21.12% for 1400 °C sintered scaffolds. For both measures, no significant differences were observed. Only a weak trend of a decrease in the bony regenerated area with an increase in sintering temperature was recognized but failed to reach a *p*-value below 0.05 with the Jonckheere-Terpstra trend test. Altogether, these results suggest that microporosities in the range between 0 and 46% have no significant effect on osteoconductivity determined by bony bridging or bony regeneration.

### 3.4. Microporosity and Compression Strength of Partial Sintered Scaffolds in Light of Osteoconduction

The microporosity of HA-based scaffolds has a minor effect on osteoconductivity, measured by the extent of bony bridging (Figure 3a), since the slope associated with the trend line for HA was 0.299 as compared with the slope associated with the trend line for TCP-based scaffolds of 1.03. Therefore, in contrast to HA-based scaffolds with high osteoconductivity over a wide range of microporosities, only highly microporous TCP-based scaffolds showed high osteoconductivity. All data for TCP-based scaffolds with identical microarchitecture were reported earlier [10]. The compression strength of sintered scaffolds was determined with the same microarchitecture used later on to perform the in vivo osteoconduction tests.

With an increase in sintering temperature, the compression strength of HA-based scaffolds increased from 0.24 ± 0.04 N/mm^2^ (mean ± S.E.M.) at 1100 °C, 0.95 ± 0.28 N/mm^2^ at 1200 °C, and 2.58 ± 0.61 N/mm^2^ at 1300 °C to 5.69 ± 1.70 N/mm^2^ at 1400 °C; only the HA-based scaffolds sintered at 1300 and 1400 °C matched the naturally occurring range of cancellous bone (Figure 3b) [34] of 2–12 N/mm^2^. In both cases, their osteoconductivity exceeded by far the osteoconductivity of TCP-based scaffolds, reaching the compression strength of cancellous bone.

### 3.5. Ion Release from Partially Sintered Scaffolds

To study the effect from the changes in the surface of the scaffolds (Table 1), we determined Ca^2+^ ions release from our four scaffold types (Figure 4a). The Ca^2+^ ion release, over 60 days, was 7.66 ± 0.32 µmol/g for scaffolds sintered at 1100 °C, 7.08 ± 0.35 µmol/g for scaffolds sintered at 1200 °C, 8.96 ± 0.58 µmol/g for scaffolds sintered at 1300 °C, and 9.33 ± 0.62 µmol/g for scaffolds sintered at 1400 °C, respectively. From the 20-day time point on, all dissolution curves were parallel to each other, suggesting that the Ca^2+^ dissolution rates from all four scaffold types were the same.

### 3.6. Osteoclastic Resorption of HA-Based Scaffolds In Vitro

The overall degradation of the scaffold is the sum of dissolution determined by ion release (Figure 4a) and cell-based resorption. To that end, first, we looked at the influence of sintering temperature on osteoclastogenesis. Scanning electron microscopy revealed that in the presence of RANKL, osteoclasts could be found on the surface of HA-based scaffolds sintered at 1100, 1200, 1300, and 1400 °C (Figure 4b). In scaffolds sintered at 1300 and 1400 °C, more osteoclasts were visible on the surface along with several non-differentiated RAW264 cells. The lower number of osteoclasts in 1100 and 1200 °C sintered scaffolds could be due to the increased penetration of the cells into the scaffolds due to higher microporosity of the material. A phenomenon, we already experienced with TCP-based scaffolds [10].

Moreover, incubation of the scaffolds seeded with RANKL-stimulated RAW264 cells to induce osteoclast formation (Figure 4b), affected the original surface of all scaffold types (Figure 5a) in terms of the appearance of resolution pits (Figure 5b,c). The most impressive observation was the effect on the surface morphology and the number of resorption pits for scaffolds sintered at 1300 °C and exposed to osteoclastic cells (Figure 5b,c) as compared with surfaces not being exposed (Figure 5a).

## 4. Discussion

Following up on our study on microporosity of 3D-printed TCP-based scaffolds to influence bone formation and osteoconduction in vivo and osteoclastic resorption in vitro [10], in this study, we assessed the same aspects with 3D-printed HA-based scaffolds. Microporosity and other measures of the nanoarchitecture were tuned by the maximum sintering temperature (Table 1). The results of our study revealed that osteoconduction and bone regeneration, of a cranial defect treated with scaffolds of identical wide-open porous microarchitecture, were at a high level for all scaffold types. However, no significant difference between peak sintering temperatures from 1100 to 1400 °C and microporosities between 0 and 46% could be detected (Figure 2). As compared with TCP-based scaffolds, the HA-based scaffolds showed a consistently higher osteoconductivity. Therefore, microporosity, as such, has no significant influence on osteoconduction of wide-open porous scaffolds. Additionally, on the surface of all types of these HA-based scaffolds, osteoclasts formed, and osteoclastic degradation was detectable.

The nanoarchitectural features of ceramics comprise parameters such as grain size, micropore diameter, and overall microporosity. They can be tuned, however, not independently off each other, by the choice of the sintering temperature [24,35] (Table 1). For HA-based scaffolds, it was shown that, at an overall porosity of 80%, an increase in microporosity from 10 to 20% was sufficient to enhance in vivo bone formation [36]. Free-formed HA-based scaffolds produced by a lost-wax methodology with or without 22% microporosity showed only a trend but no significant increase in bone ingrowth into pores of 0.35 mm [37]. There was no effect of 3–29 vol.% microporosity on either osseointegration or osteoconductivity by HA-based scaffolds implanted in rabbit femur for 8 to 12 weeks was seen by others [38]. Our scaffolds covered an even wider range of microporosity from zero to 46% and no significant influence on osteoconductivity and only a trend in bony regenerated area could be detected. Our results suggest that, for HA-based scaffolds, bone formation is independent of microporosity in the range between 0 and 45% (Figure 3 and Table 1). This might be due to the fact that we used 3D-printed wide-open porous microarchitectures where nutrient supply and waste exchange are no limiting factor for bone formation, as compared with foamed scaffolds [36] where macropore interconnectivity is limited, and thus, nutrient supply and waste exchange is heavily dependent on microporosity.

An increase in sintering temperature not only affects microporosity but also leads to phase transformations [39]. At a 2 h sintering period at 1300 °C, which is also recommended by the manufacturer of the slurry which we used here, 100% of the HA-powder remains in the HA-phase, which drops to 92% when sintered at 1400 °C [40]. High osteoconductivity was observed with sintering temperatures between 1100 and 1300 °C and microporosities between 18 and 46%. Since osteoconductivity drops only slightly with scaffolds sintered at 1400 °C, possible phase transformation has, if at all, only a minor effect on osteoconductivity or osteoclastic degradability. However, additional studies, beyond the scope of this project, would be needed to address these effects in more detail.

TCP-based scaffolds with microporosities between 10 and 25% performed in vivo equally well [24]. For silicate-substituted calcium-phosphate-based scaffolds, a minimum microporosity of 39% was needed to increase bone regeneration [41]. The same applies to wide-open porous 3D-printed TCP-based scaffolds [10] where 39% microporosity yielded a significant increase in bony bridging and bony regeneration, as compared with scaffolds with identical microarchitecture but only 0 or 22% microporosity. For both materials, a microporosity level of 39% enhanced bony regeneration. This could be due to an increased surface, facilitating an increase in protein adsorption [36], an increase in ionic solubility [42,43], and an increase in attachment points for osteoblasts [44]. In the case of our wide-open porous microarchitectures, bone forms predominantly between the material [9]. Therefore, the effect of microporosity by increased protein absorption and attachment points for osteoblasts should be very small, leaving the optimal ion dissolution as the major and indirect cause for the increase in osteoconductivity of TCP-based scaffolds with 39% microporosity [10].

Considering studies dealing with microporous ceramics, it is difficult to compare when design, material chemistry, and processing techniques vary [36,38]. In combination with our earlier study on TCP-based scaffolds [10], here, we compare TCP- and HA-based scaffolds with the identical macro- and microarchitecture in a microporosity range between 0 and 40% undergoing the same processing technique (Figure 3) and tested in the identical in vivo situation. The most striking result was that the HA-based scaffolds were superior to TCP-based scaffolds in terms of osteoconductivity (Figure 3a); only at a high microporosity of 39%, which is associated with poor mechanical strength (Figure 3b) [45], TCP-based scaffolds reached the high osteoconductivity level of HA-based scaffolds. Therefore, to produce wide-open scaffolds with high mechanical strength in combination with high osteoconductivity, HA appears to be superior to TCP.

The reason to use TCP for bone substitutes is the faster degradability [46] even though bone tissue is predominantly composed of HA [47]. Following the healing process of the scaffold-treated defect by bony bridging, a gradual degradation of the scaffold is needed for creeping substitution. To that end, we studied the degradation of our scaffolds by chemical dissolution (physicochemical degradation) (Figure 4a) and biological resorption (cellular degradation by osteoclasts) (Figure 5). The Ca^2+^ dissolution curves from the HA-based scaffolds (Figure 4a) went in parallel, suggesting that after the initial 10 days, dissolution of Ca^2+^ ions per gram of scaffold was similar for all sintering temperatures tested.

Our overall strategy in scaffold-based tissue engineering aims at a fast ingrowth of bone into the scaffold to achieve the quickest possible bony bridging of the defect [9]. Due to the coexistence of scaffold and newly formed bone in the regenerated area, the cellular degradability of the bony integrated scaffold would facilitate a natural bone turnover of the bone/scaffold composite. For all tested sintering temperatures, osteoclasts formed on the surface of the scaffolds (Figure 4b) and osteoclastic resorption visualized by resorption pits was present (Figure 5). Therefore, all HA-based scaffold types could undergo osteoclastic resorption in line with the natural bone turnover of the defects’ bony bridging.

For biphasic calcium phosphate (BCP), it has been reported that microstructural dimensions were critical in promoting osteoclastogenesis [48]. BCP-based samples with a grain diameter of 1.2 µm instead of 3.5 µm favored RAW264 cell proliferation and activity. An increase in grain size leads to a decrease in osteoclastic resorption of TCP [49]. We recently supported this finding, as scaffolds made from TCP and grain size of 3.08 µm were less susceptible to osteoclastic resorption than those with a grain size of 1.24 µm [10]. For HA in a submicron range, the grain size of 0.5 µm impaired osteoclastic formation and function as compared with scaffolds with grain size of 0.1 µm [50]. Here, we studied HA-based samples with grain sizes in the micron range between 1.07 and 5.98 µm (Table 1) and saw osteoclast formation and activity on all the surfaces and all grain sizes. Taken together, an increased grain size tends to hamper osteoclastic degradability for diverse calcium phosphates, but grain size alone might not be sufficient to predict degradability by osteoclasts, and scaffold characterization should always include an experimental test of this parameter.

## 5. Conclusions

Previously, we documented the positive influence of microporosity on osteoconduction of TCP-based scaffolds; here, we determined the effect of microporosity on osteoconduction and creeping substitution of HA-based scaffolds. In contrast to TCP-based scaffolds, with HA-based scaffolds, we found high osteoconductivity irrespective of the level of microporosity. Since microporosity is tightly linked to mechanics, HA-based scaffolds appear to be better suited for mechanically demanding bone regeneration procedures. Moreover, microporosity between 0 and 46% imposed no major effect on osteoclast formation and osteoclastic resorbability. Therefore, microporosity, as such and as shown with HA-based scaffolds, does not affect osteoconductivity or prevent osteoclastic resorbability. Together, these findings are likely to guide the choice of material for future developments in 3D-printed personalized calcium phosphate-based scaffolds.

## Figures and Tables

**Figure 1 materials-15-01433-f001:**
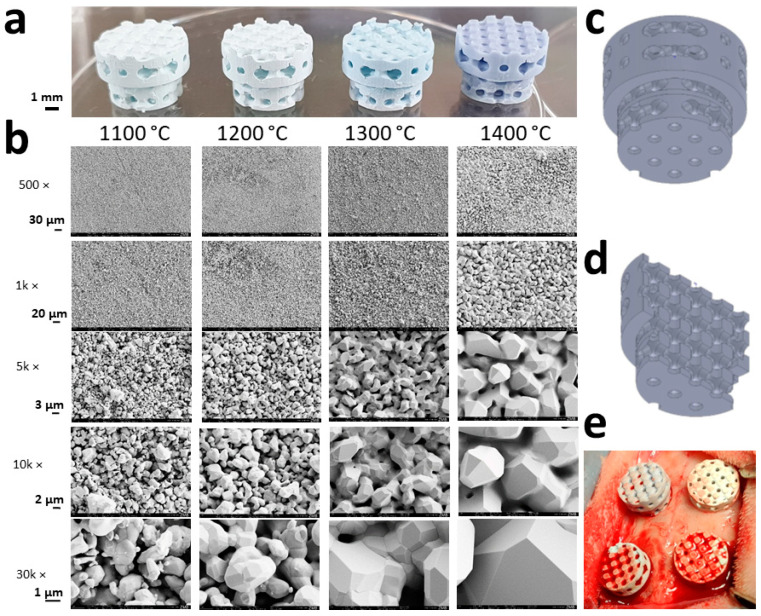
Macrographies and SEM images of the experimental scaffolds: (**a**) Scaffolds, which underwent different sintering temperatures, are shown to maintain their macro- and microarchitecture; (**b**) sintering temperatures applying to panel a and b are provided. SEM micrographs from the respective scaffolds are displayed with scales to the left; (**c**) the macroarchitecture of the full scaffold; (**d**) the macroarchitecture of the halved scaffold; (**e**) a picture taken after placement of all four different scaffolds in the calvarial bone defects of a rabbit is provided.

**Figure 2 materials-15-01433-f002:**
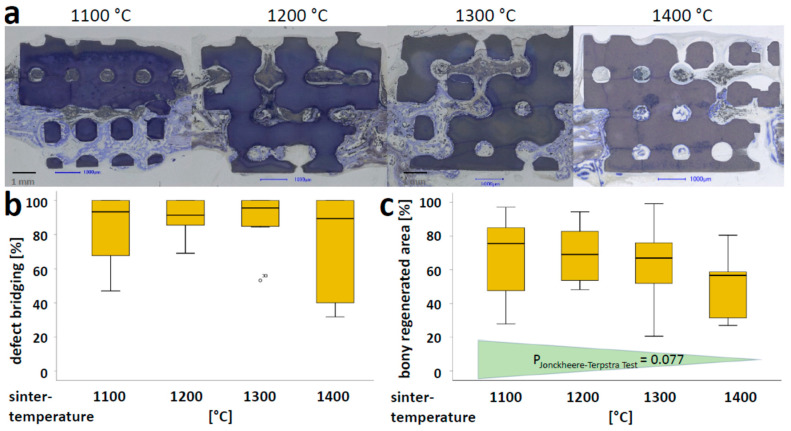
Microporosity-dependent osteoconduction and bone regeneration: (**a**) Histological sections from the middle of the noncritical-size defects treated with scaffolds sintered at 1100, 1200, 1300 or 1400 °C. Histological section from 4 weeks postoperatively is shown. Scale bars represent 1 mm. Bone (grayish purple to purple) and HA (dark bluish to black) are visualized; (**b**) defect bridging and (**c**) the formation of new bone, are displayed according to [10]. For bony regenerated area (**c**), a weak trend of a decrease in bony regenerated area with increased sintering temperature was recognized. The *p*-value of the Jonckheere–Terpstra trend test is provided.

**Figure 3 materials-15-01433-f003:**
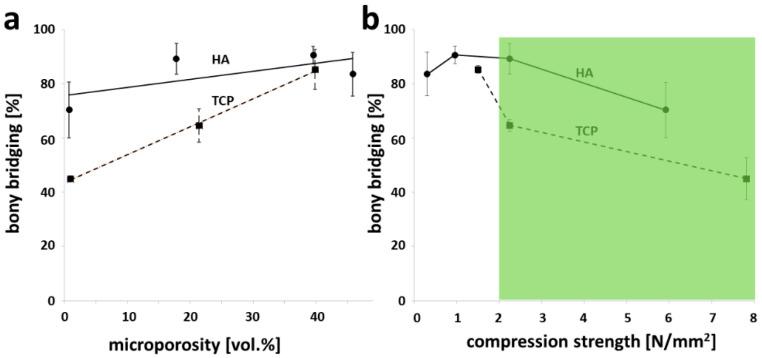
Comparison of osteoconductivity in relation to microporosity and compression strength between otherwise identical HA- and TCP-based scaffolds. The results for TCP-based scaffolds produced with the identical stl-file were generated and reported earlier [10]. (**a**) Osteoconductivity related to microporosity of test samples; (**b**) osteoconductivity related to compression strength of the partially sintered scaffolds. The values are displayed as mean ± standard deviation. The range for cancellous bone (2–12 N/mm^2^) depicted in the green shaded area was taken from [34].

**Figure 4 materials-15-01433-f004:**
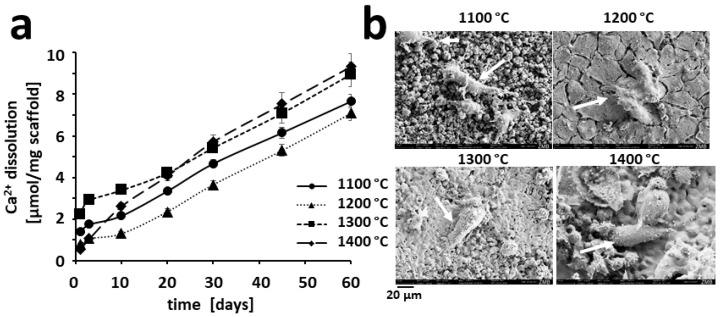
Sintering-temperature dependent Ca^2+^ ion release and osteoclast formation: (**a**) Ca^2+^ dissolution over 60 days was determined for all four scaffold types; (**b**) osteoclastogenesis from osteoclastic RAW264 cells stimulated with RANKL was studied by scanning electron microscopy. On top of all scaffolds sintered from 1100 to 1400 °C, osteoclasts could be identified. The sintering temperature, white arrows to mark osteoclasts, and a scale is provided.

**Figure 5 materials-15-01433-f005:**
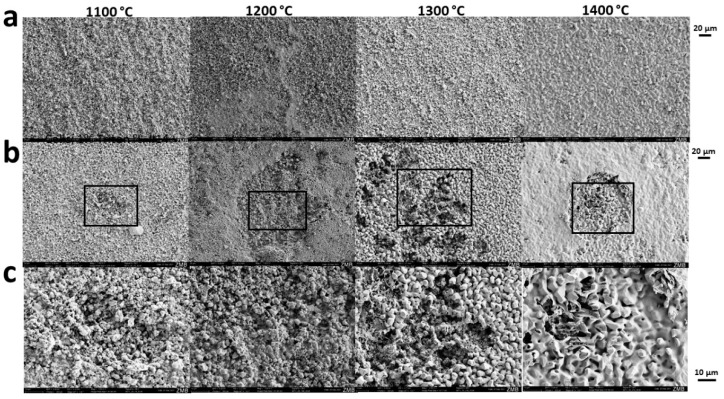
Sintering-temperature dependent osteoclastic degradation of scaffolds. Scanning electron microscopy was performed from scaffolds not exposed to cells (**a**) or seeded and exposed to osteoclastic RANKL-stimulated RAW264 cells (**b**,**c**). The maximal sintering temperature is provided at the top of each column. Scales of each panel are provided.

**Table 1 materials-15-01433-t001:** Characteristics of the sintering temperature dependent nanoarchitectures.

Peak Sinter Temperature (°C)	Grain Size (µm)	Micropore Diameter (µm)	Surface (m^2^/g)	Microporosity by Shrinkage (%)	Microporosity by Infiltration (%)
1100	1.07 ± 0.46	1.17 ± 0.47	0.79	42.48	45.85 ± 0.39
1200	1.45 ± 0.58	1.25 ± 0.32	0.57	35.92	39.58 ± 0.39
1300	3.01 ± 1.16	1.21 ± 0.50	0.34	15.00	17.79 ± 0.78
1400	5.98 ± 1.11	1.67 ± 0.52	0.24	0.00	0.74 ± 1.87

## Data Availability

The raw/processed data required to reproduce these findings cannot be shared at this time as the data also forms part of additional ongoing studies.

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
