# Peer review of "3D-Printed HA-Based Scaffolds for Bone Regeneration: Microporosity, Osteoconduction and Osteoclastic Resorption"

_materials, 2022, doi:10.3390/ma15041433_

Round 1
Reviewer 1 Report
The paper deals with 3D-printing process of bone substitutes. Special attention is paid to the effect of the temperature in the resulting nanoarchitecture. A sample of substitute specimens is used for the empirical study, involving in vivo testing in rabbits and additional in vitro tests. The paper claims that the major complications associated with these bone substitutes can be prevented using the proposed methodology. In general terms, the article seems novel, meaningful, clear and well edited.
In table 1 it is curious that sd in “microporosity by infiltration” becomes larger precisely as the mean decreases that would give a coefficient of variation CV = mean/sd that spikes upward. Please explain in some way this growing CV.
In figure 2a, it seems that the second and fourth images are the same which must be an error (it seems unlikely that they are two indistinguishable images).
In Figure 3b, the compression strength [N/mm^2] is mentioned but it is not clear what effective area was calculated to determine this stress. The shape of the implants shown in Figure 1a, appears to have holes so it is difficult to know what area would be used.
It is also mentioned that the compression curves were analyzed with Zwick Roell software, but they are not shown and it is difficult to judge such test. Comparative data with expected strengths in other types of alternative scaffolds are also not given.
The conclusions do not seem to be well connected with the rest of the argumentation within the paper. For example, the statement made at the beginning that the selected scaffold prevents non-union (the major complication of a scaffold) does not seem to be mentioned. non-union (the major complication of a scaffold) does not seem to be mentioned. Similarly, the statement in the conclusions that "since with wide 463 open-porous HA-based scaffolds" a high mechanical strength can be achieved cannot be correctly evaluated, what would be a high strength? what would be a small strength? what would be the strength of alternative systems?
Reviewer 2 Report
The manuscript studies the effect of microporosity on HA 3D printed scaffolds on the bone regeneration activity both in vitro and in vivo.
The topic is of high interest, both for the use of 3D printing technology that allows to tailor the scaffold architecture at the macro and micro level, and for the study of microporosity.
However, I found two main points that need to be addressed by the authors, in order to give to the manuscript the scientific consistency necessary for publishing.
- Authors claim to study the effect of microporosity on bone regeneration, but microporosity is not properly quantified. The method used to measure microporosity (volumetric shrinkage upon sintering and distilled water uptake) do not offer a precise evaluation. Authors set to 0% the porosity of scaffold sintered at 1400°C, however this was not confirmed by SEM images (see for instance Fig.1b). Authors performed BET analyses, but they should be further extended in order to quantify the amount of micro and mesopores for instance using BJH method. Author should refer to IUPAC definition of micro, meso and macro pores. The use of term microporosity is here abused, since not clearly identified. A quantification of the total porosity, amount of open and closed porosity and amount of macro-meso and micro pores should be reported in order to properly discuss its effect on bone regeneration.
- A further point that needs a more detailed discussion is the effect of sintering temperature on HA microstructure. XRD measurements and Rietveld refinement should be performed in order to assess their effect on bone regeneration and Ca2+ dissolution kinetics. See for instance (in this study the final sintering temperature is 1200°C.
- Last but not least, the title should be revised. The TCP results belong to a previous study. Checking the ref 10, the same method is reported so as the same inconsistency in terms of definition of microporosity and measurement. I think that the emphasis on the comparison between the two studies should be reduced, at least not reported in the title.
Some other minor comments are reported in the attached documents. The experimental part should be arranged in 4 subsections: 1. Scaffolds production.2 Materials Characterization. 3 In vitro studies. 4. In vivo studies.

Reviewer 3 Report
The title suggests the comparative study of TCP-based and HA-based scaffolds, while the paper consists of results of the research carried only on HA-based scaffolds, while these results are discussed as referred to results obtained for the TCP-based scaffolds published in the previous paper from the authors. This approach is misleading. The conclusions from the paper should be supported by the results included in the same particular paper while in this case conclusions are derived from the results of two or more different papers. The title states so, and it causes the paper to be more difficult to reading. Also, the construction of the paper referred to is quite similar what raises a suspicion that it is a twin paper where similar content is presented or separately, instead of one paper. Please justify why since the conclusions on the previously published paper state that for TCP-based scaffolds microporosity appears to be essential for optimal osteoconduction, what is the rationale for another paper with the objective to evaluate the contribution of microporosity to osteoconductivity of HA-based scaffolds? Without a strong motivation for undertaking or continuing such an aspect of research, the originality/novelty of the presented manuscript is found as low.
The research design is seen proper and the methods are described adequately as they look as repeating the methods included in the already published paper, however, results would be clearer if presented as rounded to significant digits.
Please explain how the compression strength of partially sintered scaffolds affects the osteoconductivity? There are empirical relationships allowing to estimate the mechanical properties from knowing the porosity or density of cancellous bone, and thus the based on such an approach the microporosity and compression strength should be consistent. The direct impact of the compression strength on osteoconduction remains unclear for the reader.
There is a lack of consistency between the goal specified at the end of the Introduction and conclusions derived from the results that refer mostly to the nanoscale features. This after reading the convoluted discussion leads the reader at the end with an impression that it is unknown the true findings and value of the manuscript.
This is found as the main issue with the submitted manuscript because the methods applied and research performed is of high level. The conclusions are not stated explicitly enough from the presented results and it might be resulting from that authors have divided the results into two papers and try to conclude it together.
Reviewer 4 Report
Dear Authors!
You have done important research on the relationship between microporosity, osteoconduction and osteoclastic resorption. However, there are several comments for revision.
- In the introduction, you talk about the importance of pore size, porosity, channels and pore interconnection (49-50). Why are you considering only open pores further?
- You did not find with HA-based scaffolds no stringent link between microporosity and osteoconduction in wide open-porous scaffolds. (458 -459). Everywhere you show the microporosity in%, but the most important factor is the size of the pores (their diameter, length). But these characteristics are not indicated for you. You need to estimate these dimensions for your materials.
- You indicated that the BCP-based samples with a grain diameter of 1.2 µm instead of 3.5 µm promoted the proliferation and activity of RAW264 cells. (442-443). How did you show it? Would it be useful to provide data on cell adhesion and proliferation? What was the ratio of living to dead cells? Because If you have detected the release of Ca2+ ions, then other HA components, which can be potentially toxic, may have come out into the solution.
- Porous samples can have uncontrolled resorption times (by hydrolysis), which is not accompanied by the formation of bone tissue. Have you checked it out?
Round 2
Reviewer 2 Report
There are still two points that need to be addressed:
1) Measurement of porosity and discrimination between open and closed porosity. This is mainly important because it affects osteoconductivity.
2) Discussion on how sintering at different temperatures will affect phase transformations and not only porosity.
Author Response
Dear Reviewer 2. Thanks for reminding me about these two important points. I hope we addressed these points in an appropriate way, so that the present manuscript can be published soon.
1) Measurement of porosity and discrimination between open and closed porosity. This is mainly important because it affects osteoconductivity.
We have added the numbers for the size of the micropores in table 1.
And showed that the microporosity is almost exclusively (far above 86%) open-porous.
The maximal microporosity determined by infiltration of samples sintered at 1000 °C with water was 46.43 ± 0.35 %. According to the manufacturer, the volume percentage of the binder in the slurry is 54%. That means that at least 86% if not close to 100% of the microporosity is open porous. Moreover, if the volume of HA in the scaffolds is calculated according to its specific weight and the microporosity determined by the volume of water infiltration as before, the numbers match the numbers displayed in table 1. Therefore, due to the homogeneous dispersion of the HA particles in the binder system, debinding yields in the formation of an almost exclusive open-porous microporosity where water and water-soluble substances can easily pass through.
2) Discussion on how sintering at different temperatures will affect phase transformations and not only porosity.
We have adressed this point in the discussion
An increase in sinter temperature not only affects microporosity but also leads to phase transformations [39]. At a 2 h sinter period at 1300 °C, which is also recommended by the manufacturer of the slurry which we used here, 100 % of the HA-powder remains in the HA-phase to drop to 92 % when sintered at 1400 °C [40]. High osteoconductivity we saw with sinter temperatures between 1100 - 1300 °C and microporosities between 18 and 46%. Since osteoconductivity drops only slightly with scaffolds sintered at 1400 °C, possible phase transformation have, if at all, only a minor effect on osteoconductivity or osteoclastic degradability. However, additional studies, beyond the scope of this project, would be needed to address these effects in more detail.

Reviewer 3 Report
Referring my comments the revised manuscript has been sufficiently improved to be accepted to publish.
Author Response
Dear Reviewer 3. Thanks a lot for your positive comment and the acceptance of the manuscript.
Referring my comments the revised manuscript has been sufficiently improved to be accepted to publish.
